# 3.3-Å resolution cryo-EM structure of human ribonucleotide reductase with substrate and allosteric regulators bound

Edward J Brignole[1,2†], Kuang-Lei Tsai[3†‡], Johnathan Chittuluru[3], Haoran Li[4], Yimon Aye[4§], Pawel A Penczek[5], JoAnne Stubbe[2,4*], Catherine L Drennan[1,2,4*], Francisco Asturias[3#*]

[1]Howard Hughes Medical Institute, Massachusetts Institute of Technology, Cambridge, United States; [2]Department of Biology, Massachusetts Institute of Technology, Cambridge, United States; [3]Department of Integrative Computational and Structural Biology, The Scripps Research Institute, La Jolla, United States; [4]Department of Chemistry, Massachusetts Institute of Technology, Cambridge, United States; [5]Department of Biochemistry and Molecular Biology, The University of Texas-Houston Medical School, Houston, United States

*For correspondence:
stubbe@mit.edu (JAS);
cdrennan@mit.edu (CLD);
Francisco.Asturias@ucdenver.edu
(FA)

†These authors contributed equally to this work

Present address: ‡Department of Biochemistry and Molecular Biology, The University of Texas-Houston Medical School, Houston, United States; §Department of Chemistry & Chemical Biology, Cornell University, Ithaca, United States; #Department of Biochemistry & Molecular Genetics, University of Colorado Anschutz Medical School, Denver, United States

Competing interests: The authors declare that no competing interests exist.

**Abstract** Ribonucleotide reductases (RNRs) convert ribonucleotides into deoxyribonucleotides, a reaction essential for DNA replication and repair. Human RNR requires two subunits for activity, the α subunit contains the active site, and the β subunit houses the radical cofactor. Here, we present a 3.3-Å resolution structure by cryo-electron microscopy (EM) of a dATP-inhibited state of human RNR. This structure, which was determined in the presence of substrate CDP and allosteric regulators ATP and dATP, has three $\alpha_2$ units arranged in an $\alpha_6$ ring. At near-atomic resolution, these data provide insight into the molecular basis for CDP recognition by allosteric specificity effectors dATP/ATP. Additionally, we present lower-resolution EM structures of human $\alpha_6$ in the presence of both the anticancer drug clofarabine triphosphate and $\beta_2$. Together, these structures support a model for RNR inhibition in which $\beta_2$ is excluded from binding in a radical transfer competent position when α exists as a stable hexamer.
DOI: https://doi.org/10.7554/eLife.31502.001

## Introduction

Ribonucleotide reductase (RNR), an essential enzyme in all organisms, catalyzes the reduction of ribonucleotides into deoxyribonucleotide precursors for replication and repair of DNA. Because RNR is vital for cell proliferation and genome maintenance, drugs that target human RNR are used against some of the most aggressive and challenging to treat cancers, including refractory lymphoblastic leukemia, metastatic ovarian and pancreatic cancers, and melanoma (*Aye et al., 2015*). Human RNR is a class Ia RNR, represented by eukaryotes and some prokaryotes that includes the well-studied homologs from *Escherichia coli* and *Saccharomyces cervisiae* (*Cotruvo and Stubbe, 2011*; *Brignole et al., 2012*; *Hofer et al., 2012*; *Minnihan et al., 2013b*). In the class Ia RNRs two homodimeric subunits work together to reduce diphosphate forms of all four canonical ribonucleotides (NDPs) to their deoxyribonucleotide counterparts (*Figure 1A–D*) (*von Döbeln and Reichard, 1976*; *Brignole et al., 2012*; *Hofer et al., 2012*). The smaller β subunit houses a stable diferric-tyrosyl radical cofactor generated by oxidation of a di-iron cofactor (*Atkin et al., 1973*; *Sjöberg et al., 1978*; *Larsson and Sjöberg, 1986*; *Bollinger et al., 1991*; *Cotruvo and Stubbe, 2011*) (*Figure 1C, D*). The larger α subunit contains the active site and two allosteric regulatory sites (*Brown and Reichard, 1969b*; *von Döbeln and Reichard, 1976*; *Eriksson et al., 1997*) (*Figure 1C,D*). Long-

**eLife digest** Cells often need to make more DNA, for example when they are about to divide or need to repair their genetic information. The building blocks of DNA – also called deoxyribonucleotides – are created through a series of biochemical reactions. Among the enzymes that accomplish these reactions, ribonucleotide reductases (or RNRs, for short) perform a key irreversible step.

One prominent class of RNR contains two basic units, named alpha and beta. The active form of these RNRs is made up of a pair of alpha units ($\alpha_2$), which associates with a pair of beta units ($\beta_2$) to create an $\alpha_2\beta_2$ structure. $\alpha_2$ captures molecules called ribonucleotides and, with the help of $\beta_2$, converts them to deoxyribonucleotides that after futher processing will be used to create DNA.

As RNR produces deoxyribonucleotides, levels of DNA building blocks in the cell rise. To avoid overstocking the cell, RNR contains an 'off switch' that is triggered when levels of one of the DNA building blocks, dATP, is high enough to occupy a particular site on the alpha unit. Binding of dATP to this site results in three pairs of alpha units getting together to form a stable ring of six units (called $\alpha_6$). How the formation of this stable $\alpha_6$ ring actually turns off RNR was an open question.

Here, Brignole, Tsai et al. use a microscopy method called cryo-EM to reveal the three-dimensional structure of the inactive human RNR almost down to the level of individual atoms. When the alpha pairs form an $\alpha_6$ ring, the hole in the center of this circle is smaller than $\beta_2$, keeping $\beta_2$ away from $\alpha_2$. This inaccessibility leads to RNR being switched off.

If RNR is inactive, DNA synthesis is impaired and cells cannot divide. In turn, controlling whether or not cells proliferate is key to fighting diseases like cancer (where 'rogue' cells keep replicating) or bacterial infections. Certain cancer treatments already target RNR, and create the inactive $\alpha_6$ ring structure. In addition, in bacteria, the inactive form of RNR is different from the human one and forms an $\alpha_4\beta_4$ ring, rather than an $\alpha_6$ ring. Understanding the structure of the human inactive RNR could help scientists to find both new anticancer and antibacterial drugs.

DOI: https://doi.org/10.7554/eLife.31502.002

distance radical transfer (RT) from $\beta_2$ to $\alpha_2$ generates a transient active site thiyl radical in $\alpha_2$ that catalyzes NDP reduction and reverse radical transfer from $\alpha_2$ to $\beta_2$ regenerates the tyrosyl-radical in $\beta_2$ on every turnover (*Licht et al., 1996*; *Minnihan et al., 2013b*) (*Figure 1A*). This remarkable inter-subunit RT minimally requires formation of an $\alpha_2\beta_2$ subunit complex (*Brown and Reichard, 1969a*; *Rofougaran et al., 2006*; *Rofougaran et al., 2008*).

Crystal structures of individual dimeric subunits from *E. coli*, yeast, mouse, and human RNRs have revealed general structural similarity (*Nordlund et al., 1990*; *Uhlin and Eklund, 1994*; *Voegtli et al., 2001*; *Strand et al., 2004*; *Xu et al., 2006*; *Smith et al., 2009*; *Fairman et al., 2011*). Differences include a conspicuous three-helix insert in eukaroytic $\alpha$ (*Xu et al., 2006*) that is absent in *E. coli* (*Figure 1C,D*). Additionally, the C-termini of $\alpha_2$ and $\beta_2$ have not been detected in the available crystal structures, due to flexibility essential to their function. The unstructured C-terminal $\beta_2$ tail interacts with a specific binding site on $\alpha_2$ and participates in inter-subunit RT. The $\alpha_2$ C-terminus emanates from a strand in the active site bearing two redox active residues on the RT pathway (Tyr737 and Tyr738 in human $\alpha$, equivalent to Tyr730 and Tyr731 in *E. coli* $\alpha$), which becomes disordered shortly following these Tyr residues, preventing direct observation of a pair of Cys residues that are required for shuttling reducing equivalents to the active site (*Figure 1C,D*).

A species-specific combination of allosteric, transcriptional, post-translational, and subcellular locational controls regulates RNR activity to maintain appropriate concentrations and proportions of deoxyribonucleotides and ensures fidelity of DNA synthesis and repair (*Hofer et al., 2012*; *Guarino et al., 2014*). Allosteric regulation depends on two nucleotide-binding sites in $\alpha_2$ that, through modulation of $\alpha_2$ conformation and subunit oligomerization, tune activity and substrate specificity, as described below.

The allosteric binding site that regulates substrate preference is located at the $\alpha$-dimerization interface on one face of a conserved structural element, called loop 2, whose opposite face makes contacts with the substrate base (*Figure 1C,D*). dATP binding to this effector site results in a preference for CDP and UDP reduction; TTP for GDP reduction; and dGTP for ADP reduction (*Brown and*

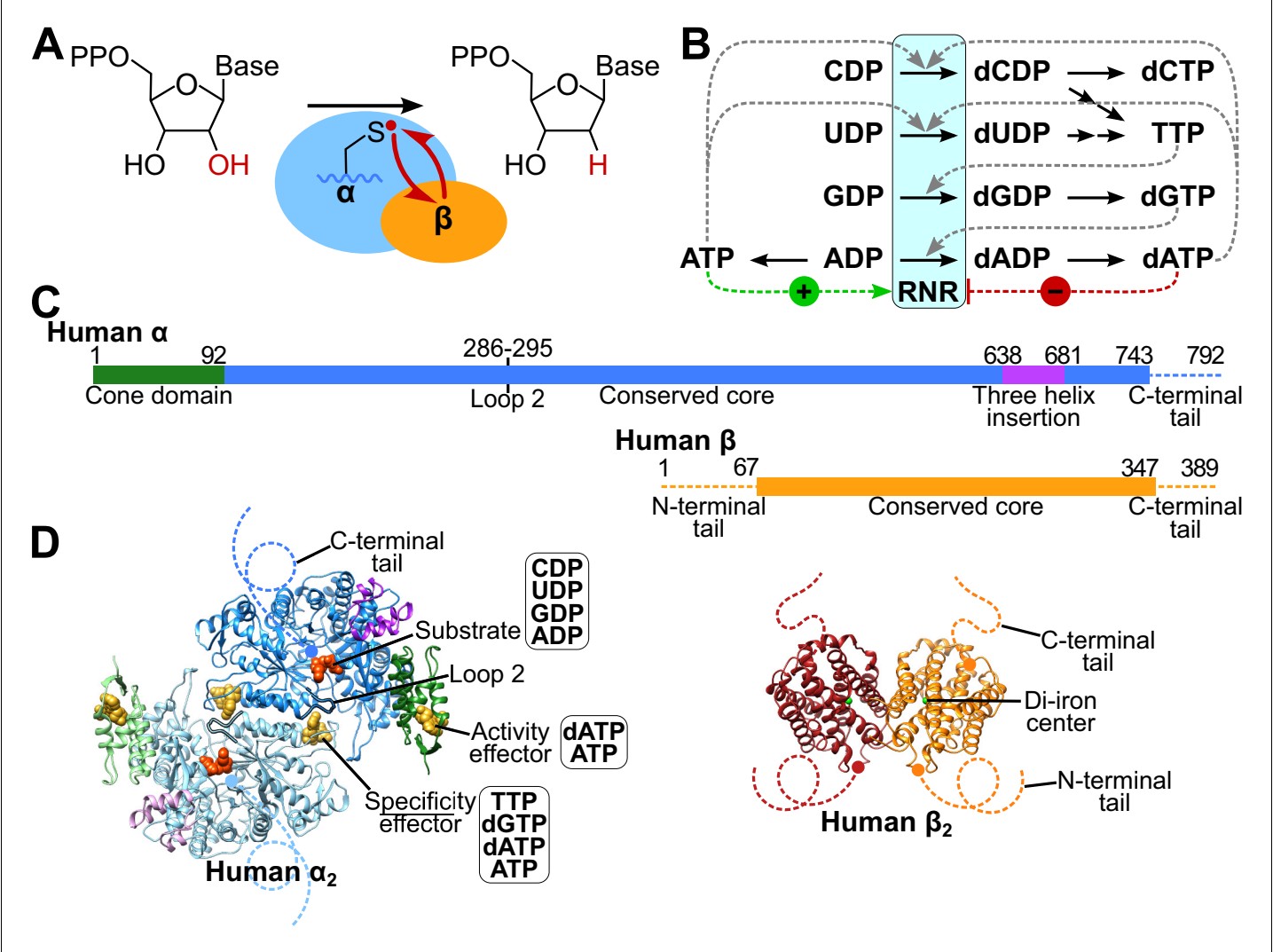

**Figure 1.** Structure and regulation of ribonucleotide reductase. (**A**) Reaction of class Ia RNR. (**B**) Diagram of allosteric regulation of activity and substrate specificity by ATP and deoxyribonucleoside triphosphate (dNTP) effectors. The blue box represents RNR catalyzing NDP reduction (black arrows). Downstream black arrows indicate further processing by other enzymes to dNTPs. The dNTP products of the pathway feed back to alter the preference of RNR toward the indicated substrate (dashed grey lines) and promote or inhibit overall activity (green and red dashed lines, respectively). (**C**) Schematics of human α and β subunits highlighting major structural features. Unstructured N- and C-termini are indicated by dashed lines. (**D**) Structures of the human α and β homodimers. Subunits of $\alpha_2$ (from our $\alpha_6$ cryo-EM structure) are colored as in panel **C** with one subunit in faded colors. Bound CDP substrate (orange) and dATP effectors (yellow) are shown as spheres. Subunits of $\beta_2$ (PDB: 2UW2) are orange and red, iron atoms in green. Dashed lines indicate unstructured termini.

DOI: https://doi.org/10.7554/eLife.31502.003

*Reichard, 1969b*; *von Döbeln and Reichard, 1976*) (*Figure 1B*). Recent crystal structures of *E. coli* RNR with all four specificity effector-substrate pairs (dATP-CDP, dATP-UDP, TTP-GDP, and dGTP-ADP) reveal the molecular basis of specificity regulation in this archetypal class Ia RNR (*Zimanyi et al., 2016*). Although the rules of allosteric regulation of specificity (*Figure 1B*) appear to be conserved, whether the molecular features of specificity regulation are the same in eukaryotic RNRs remains an open question (*Xu et al., 2006*; *Ahmad et al., 2012*; *Zimanyi et al., 2016*).

Overall RNR activity can also be allosterically regulated. For class Ia RNRs, dATP binding to the cone domain at the N-terminus of $\alpha_2$ (*Figure 1C,D*) has an inhibitory effect, whereas ATP binding reverses this inhibition (*Brown and Reichard, 1969b*; *Rofougaran et al., 2006*; *Rofougaran et al., 2008*). In the prototypical system from *E. coli*, binding of the allosteric inhibitor dATP within the

cone domain results in an $\alpha_4\beta_4$ ring-shaped structure that holds the $\alpha_2$ and $\beta_2$ subunits in a configuration incompatible with RT (*Ando et al., 2011*; *Zimanyi et al., 2012*) (*Figure 2A*). ATP binding shifts the equilibrium from the inactive $\alpha_4\beta_4$ state to the active $\alpha_2\beta_2$ state (*Rofougaran et al., 2008*; *Ando et al., 2011*). Thus, activity regulation for *E. coli* RNR involves oligomeric state changes that are modulated by the ratio of dATP to ATP in the cell.

Eukaryotic RNRs do not appear to form $\alpha_4\beta_4$-ring structures (*Fairman et al., 2011*; *Aye et al., 2012*; *Ando et al., 2016*). Instead human $\alpha_2$ has the propensity to form $\alpha_6$ rings in the presence of both ATP and dATP (*Rofougaran et al., 2006*; *Fairman et al., 2011*; *Ando et al., 2016*) and the stability of the $\alpha_6$ ring appears to regulate activity (*Ando et al., 2016*) (*Figure 2B*). Rings formed with dATP are stable, showing no oligomeric state change upon addition of $\beta_2$, whereas $\alpha_6$ rings formed in the presence of ATP are unstable and disassemble into active state(s) upon addition of $\beta_2$ (*Ando et al., 2016*). Support for the idea that 'stable' $\alpha_6$ rings correspond to an inhibited form of human RNR comes from studies with clofarabine (ClF), an adenosine analog used for cancer treatment, and related analogs cladrabine and fludarabine (*Aye and Stubbe, 2011*; *Aye et al., 2012*; *Wisitpitthaya et al., 2016*). In those studies, human RNR treated with ClF diphosphate (ClFDP) or triphosphate (ClFTP) lead to inactive enzyme and the formation of extremely stable 'persistent' $\alpha_6$ rings (*Aye and Stubbe, 2011*; *Aye et al., 2012*). Structural information about $\alpha_6$ rings has been limited to a 9-Å resolution crystal structure of human $\alpha$ obtained in the presence of dATP (*Ando et al., 2016*), a 6.6-Å resolution crystal structure of yeast $\alpha$ with dATP (*Fairman et al., 2011*), and a 28-Å resolution EM structure dATP-induced $\alpha_6$ from yeast in the presence of $\beta_2$ (*Fairman et al., 2011*). These low-resolution structures were sufficient to establish the overall $\alpha_6$ subunit arrangement, but failed to reveal the molecular basis for inactivity of the $\alpha_6$ ring. Here, we use state-of-the-art cryo-EM to determine the structure of human RNR $\alpha_6$ at the near-atomic resolution of 3.3 Å and probe the mechanisms of allosteric regulation of activity and specificity for the human class Ia RNR enzyme.

## Results

### Optimal conditions for cryo-EM were established empirically using negative stain EM

Human $\alpha$ forms ring-like particles composed of three $\alpha$-dimers (*Figure 3A*) upon addition of 0.05 mM dATP (*Figure 3—figure supplement 1C,D*) as seen previously in the low-resolution crystal structures of human and yeast $\alpha$ (*Fairman et al., 2011*; *Ando et al., 2016*) and in human RNR in the presence of ClFDP (*Aye et al., 2012*) (*Figure 3—figure supplement 1F*). These $\alpha_6$ rings formed with dATP alone showed some structural flexibility. $\alpha_6$ rings were also the predominant form observed following incubation of $\alpha$ with 1 mM ATP, but some particles were incomplete or partially open rings, indicating that the subunit contacts are tenuous, consistent with the proposition that the rings formed in the presence of ATP are not stable (*Figure 3—figure supplement 1B*). Increasing ATP concentration to 3 mM resulted in an increase in the proportion of dissociated rings and free $\alpha$-dimers. At 10 mM ATP we observed fewer $\alpha_6$ rings and a filamentous form of $\alpha$ is now apparent. Analysis of the filamentous oligomers showed that they are composed of $\alpha_2$ units chained together end-to-end, often only three units in length. With TTP, the dimeric $\alpha_2$ form is present exclusively (*Figure 3—figure supplement 1A*) as seen in prior studies (*Thelander et al., 1980*; *Reichard et al., 2000*; *Rofougaran et al., 2006*; *Aye and Stubbe, 2011*; *Fairman et al., 2011*; *Scott et al., 2001*). Finally, we examined $\alpha$ in the presence of 0.05 mM dATP and 3 mM ATP, a combination expected to be physiological relevant. In dividing cells where RNR is actively producing deoxynucleotides, the dATP concentration is approximately 0.024 mM and the ATP concentration is approximately 3 mM (*Traut, 1994*), thus the combination of 0.05 mM dATP and 3 mM ATP is expected to mimic cellular effector concentrations under which this higher dATP concentration inhibits RNR. To verify that 0.05 mM dATP inhibits human RNR even in the presence of 3 mM ATP, enzyme assays were performed, which showed inactivation (*Figure 3—figure supplement 2*). Under negative stain EM, this combination of effectors (0.05 mM dATP and 3 mM ATP) yielded data sets in which $\alpha_6$ rings are preponderant and could be combined into a well-defined 3D structure with D3 point group symmetry (*Figure 3—figure supplement 1E*). Given that this combination of effectors provided the most structurally homogeneous preparation of $\alpha_6$ rings, we used it for the high-resolution cryo-EM analysis. It is unfortunate that structural stability is so sensitive to nucleotide identity and concentration,

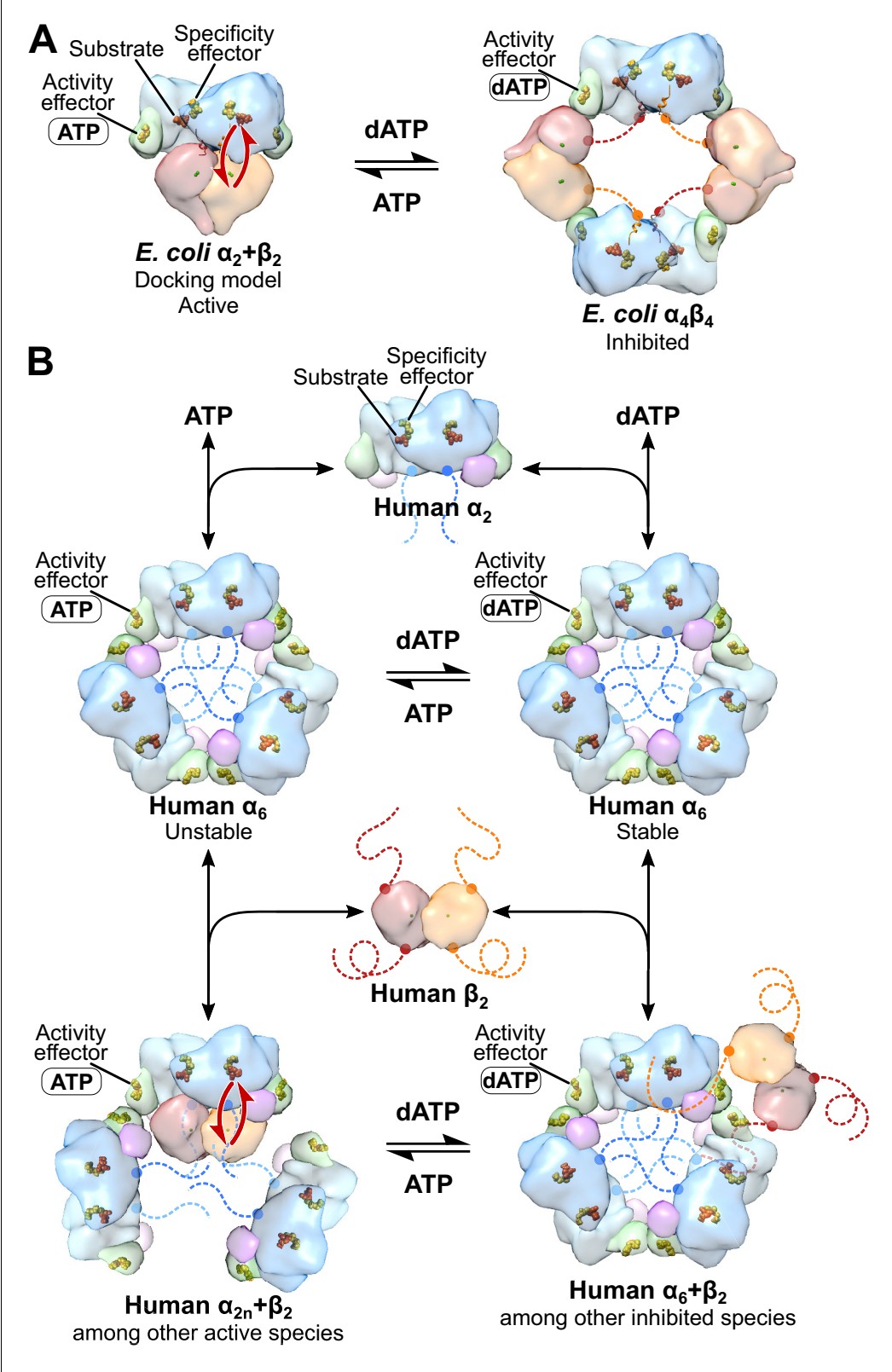

**Figure 2.** Comparison of mechanisms of allosteric regulation of activity for *E. coli* and human RNRs. (**A**) The molecular basis of allosteric regulation of activity in class Ia RNR from *E. coli*. dATP inhibits activity by binding to the activity effector site in the cone domain (green) in α and promoting conversion of the active $\alpha_2\beta_2$ complex (modeled by docking PDB: 5CNS chains A and B with PDB: 1RIB) to an inhibited $\alpha_4\beta_4$ ring (PDB: 5CNS). In particular, with dATP bound to the activity effector site in α, the cone domain forms an interface with β, leading to $\alpha_4\beta_4$ ring formation. ATP restores

*Figure 2 continued on next page*

*Figure 2 continued*

activity by displacing dATP in the activity effector site in the cone domain, which disrupts the α-β interface of the $\alpha_4\beta_4$ ring, and further pushes the equilibrium toward the $\alpha_2\beta_2$ complex. The molecular basis by which dATP promotes α-β interface formation and ATP disrupts it has not been established. Long-distance radical transfer between α and β is illustrated by red arrows. (B) The molecular basis of allosteric regulation of activity in human class Ia RNR (see text in Discussion). Briefly, $\alpha_2$ forms $\alpha_6$ in the presence of both ATP and dATP. The stability of the hexamer formed determines whether the enzyme is active when $\beta_2$ is added. dATP-induced hexamers are stable and inactive whereas ATP-induced hexamers are unstable and activated by $\beta_2$ addition. Schematic of human RNR was prepared using the same models and coloring as *Figure 1*.

DOI: https://doi.org/10.7554/eLife.31502.004

since these constraints limit the ability to determine high-resolution structures of the human RNR with all combinations of substrate and effectors. The near-atomic resolution structure described below was determined in the presence of substrate CDP and effectors dATP and ATP.

## Near-atomic resolution cryo-EM map of an inhibited $\alpha_6$ complex

We obtained a near-atomic resolution structure of the human α subunit of RNR in the presence of 0.05 mM dATP, 3 mM ATP and substrate CDP. Based on the lack of enzymatic activity under these conditions, this first near-atomic resolution structure of human $\alpha_6$ is in a dATP-inhibited state (*Figure 3A*). Imaging of the cryo-EM samples as dose-fractionated movie frames from a direct electron detector allowed for correction of specimen movement and compensation for radiation damage (*Table 1*). To eliminate concerns about model bias, particles were selected automatically, clustered using Iterative Stable Alignment and Clustering (ISAC) (*Yang et al., 2012*), and an initial 3D map was generated de novo using stochastic hill climbing refinement (*Elmlund et al., 2013*) implemented in SPARX (*Hohn et al., 2007*). The initial map was used as reference for refinement of image alignment parameters using SPARX, to obtain a final map that has an overall resolution of approximately 3.3 Å, and a maximum resolution of 3.15 Å around the α-α dimer interface (*Figure 3—figure supplements 3* and *4*). At this resolution densities can be distinguished clearly for the spiraling backbone of α-helices, the strands of β-sheets, residue side chains, and bound substrate and effector nucleotides (*Figure 3A*). The final model contains six α subunits with residues 1 through 743 of the 792 amino acids. Although both ATP and dATP were present, dATP was the nucleotide included in the final model, because it is a better fit to the density in both allosteric sites and is also expected to have higher affinity for both sites (*Brown and Reichard, 1969b*; *Reichard et al., 2000*; *Kashlan and Cooperman, 2003*). Thus, each α subunit contained one CDP in the active site, one dATP in the specificity site and one dATP in the activity site. Standard metrics of model quality show excellent statistics and fit to the cryo-EM map (*Table 1*).

## Cone-domain repositioning and hydrophobic interactions stabilize the $\alpha_6$ ring

Six α subunits, organized as three dimeric units, make contacts through their cone domains to assemble an $\alpha_6$ ring that resembles the overall arrangement seen previously in lower resolution crystal structures of dATP-inhibited human (*Ando et al., 2016*) and yeast α (*Fairman et al., 2011*) (*Figure 3A*). The $\alpha_6$ ring is ~180 Å in diameter and ~80 Å thick with a central hole that is constricted to 60 Å near each opening but widens to ~80 Å midway through the ring's interior. Each α subunit from the $\alpha_6$ ring is similar to the 2.4-Å resolution crystal structure of human $\alpha_2$ (*Fairman et al., 2011*) with an RMSD over 738 Cα atoms of 2.25 Å. Differences are localized to three regions (*Figure 3—figure supplement 5*), the cone domain, a β-hairpin loop (residues Ile624–Val637) adjacent to the cone domain, and loop 2. The largest change is rotation of the cone domain by 20°, which is necessary for the three α dimers to interact with the requisite geometry to assemble a closed $\alpha_6$ ring (*Figure 3B,C*).

Density consistent with $Mg^{2+}$-dATP is observed within the cone domain close to the subunit interface (*Figure 3D*), where the nucleotide effector makes similar contacts to those reported in crystal structures of human $\alpha_2$ and *E. coli* $\alpha_4\beta_4$ (*Fairman et al., 2011*; *Zimanyi et al., 2012*; *Zimanyi et al., 2016*). In crystal structures of *E. coli* RNR (*Zimanyi et al., 2012*, *2016*), His59 forms a hydrogen bond with the deoxyribose 3'-hydroxyl group of dATP. The equivalent residue in human α, Asp57, was not observed in the human $\alpha_2$ crystal structure with dATP bound (*Fairman et al., 2011*), but density corresponding to it is apparent in the EM map and modeling of the Asp57 side chain shows

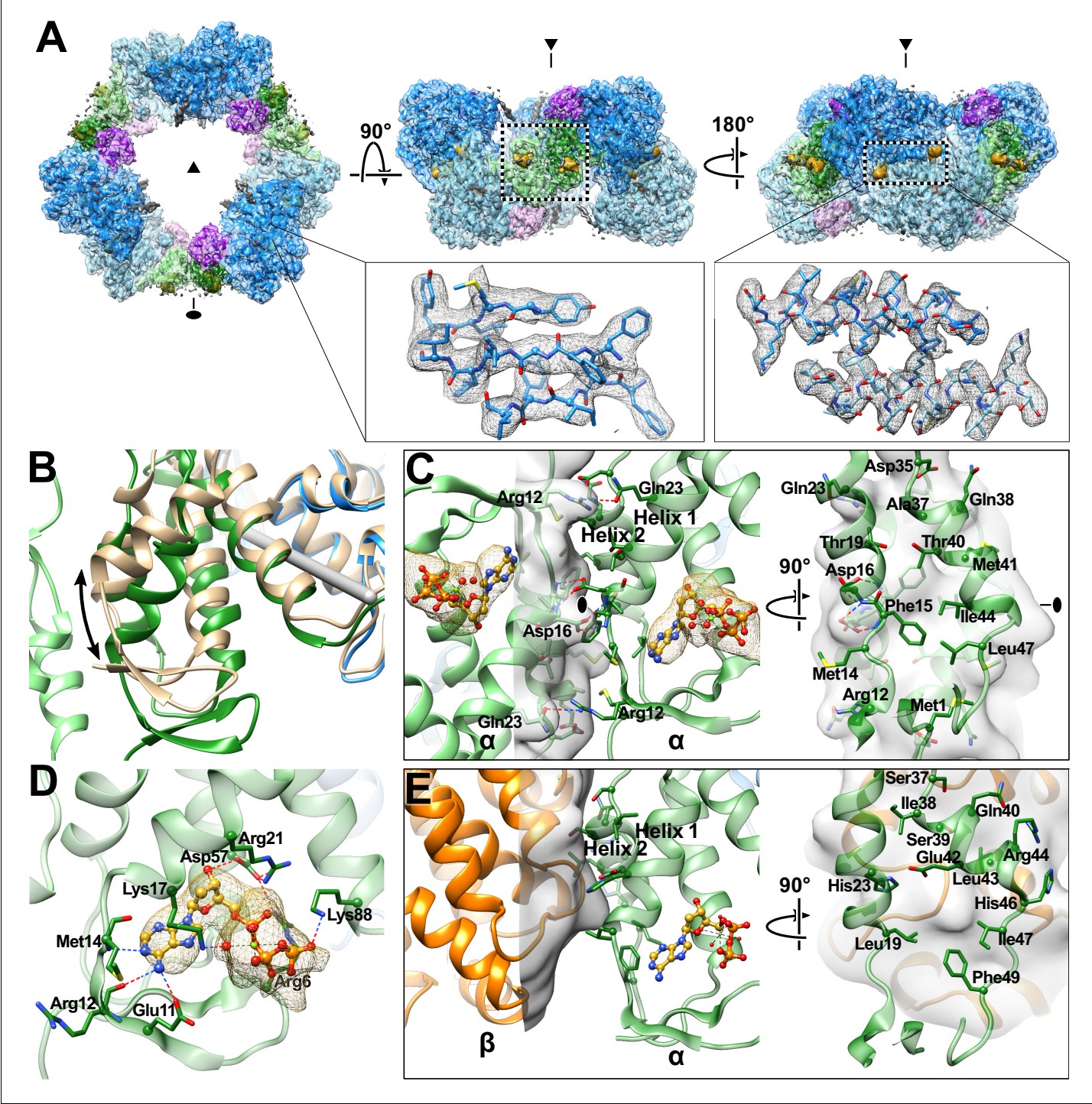

**Figure 3.** Cryo-EM density map reveals cone domain motion and contacts that assemble the human α₆ ring. (A) Human α₆ ring viewed along the three-fold (triangle) and two-fold (oval) symmetry axes. Dashed box in middle panel highlights cone domain interactions detailed in panel C. Insets show density (grey) and atomic model for a three strand segment of the catalytic core β-barrel (residues Phe297-Leu301, Leu328-Ile334, Tyr404-Tyr407) and two helices (residues Ile228-Ser244) that pair at the α dimer interface. Regions of the structure are colored as in *Figure 1* with alternating subunits in faded colors. dATP densities in yellow. (B) Cone domain in the human α₆ ring structure is ~20° away from its position in the human α₂ crystal structure colored tan (PDB: 3HNC). The gray rod indicates the rotation axis. (C) Cone domain viewed along the two-fold symmetry axis (oval) and an orthogonal view showing contacting residues, which are mostly hydrophobic. (D) dATP in the cone domain of the cryo-EM structure makes hydrogen bonds through its base, sugar, and phosphates indicated by dashed lines. Nucleotide density is shown in yellow mesh with dATP carbon yellow, oxygen red,

*Figure 3 continued on next page*

eLIFE Research article
Biochemistry and Chemical Biology | Biophysics and Structural Biology

*Figure 3 continued*

phosphorus gold, and nitrogen blue. (E) The dATP-inhibited $\alpha_4\beta_4$ ring of *E. coli* (PDB: 5CNS) uses the same face (helices 1 and 2) of the cone domain to contact the β subunit (orange). For comparison, the cone domain of *E. coli* α is oriented as in panel (C).

DOI: https://doi.org/10.7554/eLife.31502.005

The following figure supplements are available for figure 3:

**Figure supplement 1.** Abundance and definition of human $\alpha_6$ rings depends on identity and concentrations of nucleotides used.

DOI: https://doi.org/10.7554/eLife.31502.006

**Figure supplement 2.** 0.05 mM dATP is sufficient to almost completely eliminate CDP reductase activity in the presence of 3 mM ATP.

DOI: https://doi.org/10.7554/eLife.31502.007

**Figure supplement 3.** Cryo-EM analysis of the dATP-inhibited $\alpha_6$ ring.

DOI: https://doi.org/10.7554/eLife.31502.008

**Figure supplement 4.** Local resolution analysis of the cryo-EM map of dATP-inhibited $\alpha_6$ ring.

DOI: https://doi.org/10.7554/eLife.31502.009

**Figure supplement 5.** Cone domain, loop 2, and a hairpin loop have moved between human $\alpha_2$ and $\alpha_6$ ring structures.

DOI: https://doi.org/10.7554/eLife.31502.010

that its position would allow it to hydrogen-bond with dATP. The observation of an interaction between Asp57 and dATP in the human RNR EM structure is an important finding, as mutation of Asp57 to Asn results in loss of inhibition by dATP in eukaryotic RNRs (*Caras and Martin, 1988*; *Reichard et al., 2000*).

The interface between $\alpha_2$ subunits in the $\alpha_6$ ring involves cone domain helices 1 and 2 (*Figure 3C*). Remarkably, the same two helices of the cone domain make α-β contacts in the *E. coli* $\alpha_4\beta_4$ dATP-inhibited ring, although the residues involved are not conserved (*Figure 3E*). 740 Å$^2$ of solvent-accessible surface area is buried per subunit upon $\alpha_6$ ring formation, which is somewhat more extensive than the 575 Å$^2$ per subunit buried surface in the *E. coli* α-β contact (measured in 5CNS [*Zimanyi et al., 2016*]). In the human $\alpha_6$ ring structure, the $\alpha_2$-$\alpha_2$ inter-subunit contacts are primarily composed of a hydrophobic core with Phe15 and Ile44 positioned on the two-fold symmetry axes and the side chains of Met1, Ala37, Thr40, Met41, and Leu47 contributing additional contacts (*Figure 3C*). In addition to hydrophobic shape complementarity, Arg12 and Gln23 may contribute a hydrogen bond at the flanks of the interface. Asp16, a highly conserved residue that when mutated was found to disrupt $\alpha_6$ ring formation and reduce inhibition by dATP (*Fairman et al., 2011*), also sits at the two-fold symmetry axis. Unfortunately, the side chain of Asp16 has poorly defined density beyond C$_\beta$ of the side chain, making it difficult to fully explain the high degree of conservation. Overall, however, the density quality is impressive (*Video 1*), providing the first near-atomic view of the molecular interactions responsible for human RNR inhibition by dATP.

## One $\beta_2$ binds at the periphery of $\alpha_6$ rings

One explanation for why a stable $\alpha_6$ ring is inactive is that $\beta_2$ cannot access $\alpha_2$ for radical generation in this state. To evaluate the interaction between human α and β, we prepared negative stained specimens of α and β with the same mixture of ATP, dATP and CDP used to determine our cryo-EM $\alpha_6$ structure. Despite the fact that β was added in equimolar ratio with α, $\alpha_6\beta_6$ complexes did not form. Instead, we observed that only a fraction of $\alpha_6$ rings have a single, variably positioned additional density, which we attribute to a single $\beta_2$, consistent with earlier molecular mass observations of an $\alpha_6\beta_2$ complex of mouse RNR (*Rofougaran et al., 2006*). The majority of rings showed no $\beta_2$ density at all (*Figure 4A*). Similar results were obtained with CIFTP-inhibited RNR. ISAC averages that result from cryo-EM specimens prepared with ClFTP exhibit a single additional $\beta_2$ density protruding from the ring or no additional density (*Figure 4B*). Collectively, these results indicate that $\beta_2$ subunits show limited interaction affinity for stable $\alpha_6$ rings.

A 3D reconstruction of an $\alpha_6\beta_2$ complex calculated from images of ClFTP-inhibited RNR shows $\beta_2$ density only above the ring plane, not within it (*Figure 4C*). Whether $\beta_2$ density was above or within the ring was not clear from previous 2D images (*Fairman et al., 2011*). These new data allow us to establish that $\beta_2$ density is above the ring and to consider why this is the case. The dimensions of $\beta_2$ and the $\alpha_6$ ring provide a simple explanation for exclusion of $\beta_2$ from the inner ring cavity. At ~80 Å wide on its longest edge, $\beta_2$ would have difficulty navigating through the ~60 Å wide entrance to the cavity. If $\beta_2$ were able to access the interior of the ring, it would still need to be able to assume

**Table 1.** Summary of single-particle data collection, 3D reconstruction, and model refinement

| Imaging parameters and 3D reconstruction | |
| --- | --- |
| Acceleration voltage (kV) | 300 |
| Magnification (X) | 22,500 |
| Pixel size (Å) | 1.315 |
| Frame rate (s$^{-1}$) | 5 |
| Exposure time (s) | 7.6 |
| Total exposure (e$^-$ / Å) | 44 |
| Particles<br>    Micrographs used for selection<br>    Defocus range (μm)<br>    Windowed<br>    In final 3D reconstruction | <br>2144<br>0.7–3.5<br>~150,000<br>43,885 |
| Resolution<br>    'Gold-standard' at FSC 0.143 (Å) | <br>3.3 |
| **Model refinement** | |
| Resolution in phenix.real_space_refine (Å) | 3.0 |
| Number of atoms/residues/molecules<br>    NCS restrained chains<br>    Protein atoms, residues per chain<br>    Nucleotide atoms, molecules per chain<br>    Mg$^{2+}$ atoms per chain<br>    Water molecules per chain | <br>6<br>5958, 745<br>85, 3<br>2<br>6 |
| Secondary structure restraints (per chain)<br>    Helices<br>    Sheets<br>    Ramachandran<br>    Hydrogen bonds<br>    C-beta torsion restraints (per chain) | <br>29<br>7<br>743<br>252<br>1404 |
| Ramachandran angles<br>    Favored<br>    Allowed<br>    Outliers | <br>94.3<br>5.7<br>0.0 |
| r.m.s. deviations<br>    Bond lengths (Å)<br>    Bond angles (°) | <br>0.01<br>1.28 |
| Molprobity<br>    Score<br>    Clashscore<br>    Omegalyze outliers (residues per chain) | <br>1.76<br>6.89<br>1 |
| EMRinger score | 3.09 |

DOI: https://doi.org/10.7554/eLife.31502.011

an RT-competent position with respect to $\alpha_6$. Docking (based on a ~30-Å resolution EM structure of the $\alpha_2\beta_2$ state of *E. coli* RNR [*Brignole et al., 2012*; *Uhlin and Eklund, 1994*; *Minnihan et al., 2013a*]) suggests that the ability of $\beta_2$ to assume a catalytically relevant position is constrained by cone domains and the three-helix insertion motif on $\alpha$ (*Figure 4D*). Additionally, the 'cavity' of the $\alpha_6$ ring is not empty. As in all structures of RNRs, the 49-residue-long C-terminal tail of $\alpha$ is disordered and could not be directly detected in the 3.3-Å resolution cryo-EM structure. However, the locations of the last visible residues suggest that the six disordered C-terminal tails of $\alpha_6$ are likely pointing into the central cavity. Consistent with this location for the C-terminal tails, a 3D variance map shows partial densities protruding into the $\alpha_6$ ring (*Figure 4E*). Six 49-residue-long flexible tails would further impede ring access by $\beta_2$ (*Figure 2B*).

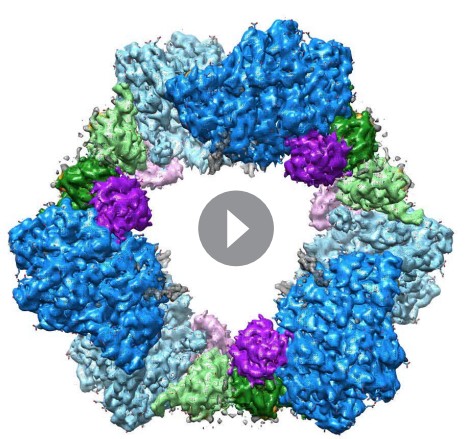

**Video 1.** Overview of the cryo-EM density and detail of the intersubunit contacts made by the cone domain to stabilize the $\alpha_6$ ring. Starting from a view down the three-fold symmetry axis, the structure rotates to a view of the cone domains on the two-fold symmetry axis, zooms into the intersubunit interface as seen in **Figure 3C**, and then rocks back and forth over a range of 30°. Density is shown as mesh colored according to regions of the structural model as in **Figure 1**.
DOI: https://doi.org/10.7554/eLife.31502.012

## Molecular basis for allosteric specificity regulation appears to be conserved for the CDP-dATP pair

No high-resolution structures of RNRs in an active state, capable of inter-subunit RT to form the catalytic thiyl radical, have been reported. However, it has not been necessary to crystallize an active state of RNR to visualize substrate and effector binding, given that substrates and effectors bind well to pre-catalytic states. In fact, substrate/effector binding increases $\alpha_2$-$\beta_2$ affinity five-fold (**Ingemarson and Thelander, 1996**; **Hassan et al., 2008**), suggesting that substrate/effector binding precedes subunit association and RT. That being said, crystal lattice contacts have historically complicated the analysis of substrate- and effector-bound structures of RNRs, resulting in poor quality density for either substrate, effector, or loop 2, the loop that communicates between the substrate and effector-binding sites (**Zimanyi et al., 2016**). Here, using cryo-EM, we notably find clear density for both substrate and effector and are able to assess their binding interactions (**Figure 5A**, **Video 2**). dATP can be modeled into the cryo-EM map at the specificity site located at the $\alpha_2$ dimer interface, where it makes contacts with loop 2 (residues 286–295) and loop 1 (residues 255–271) from the neighboring subunit (**Figure 5B**). Backbone amides of residues Ala263 and Gly264 in loop 1 establish contacts with the β- and γ-phosphates similar to those seen in crystal structures of yeast or human α with bound TTP or human α with dATP (**Xu et al., 2006**; **Fairman et al., 2011**). As previously noted for those structures, the orientation of loop 1 in the eukaryotic $\alpha_2$ is different from that in E. coli $\alpha_2$, but in both cases contacts are made by the loop to the effector phosphates. As in other RNR structures (**Larsson et al., 2004**; **Xu et al., 2006**; **Fairman et al., 2011**), Arg256 provides further stabilization of the γ-phosphate, Lys243 (not on loop 1 or loop 2) interacts with the α- and β-phosphates, and the Asp226 side chain makes hydrogen bonds with the O3' of the deoxyribose sugar.

As previously observed in E. coli, the backbone amide and carbonyl of loop 2 residue Asp287 (human numbering; Ser293 in E. coli) are involved in the specific recognition of the adenine base of effector dATP (**Zimanyi et al., 2016**), positioning loop 2 such that the side chain of the adjacent Gln288 (Gln294 in E. coli) is directed into the active site where it can hydrogen bond to the base of substrate CDP (**Figure 5A**) The presence of an extra residue in loop 2 in human RNR (compared to E. coli) results in an additional contact to the dATP adenine base made by the Gly289 carbonyl, further stabilizing a 'Gln-in' conformation of loop 2. Since the presence of the Gln side chain in the active site would inhibit the binding of the larger purine bases, the "Gln-in "position confers preference for pyrimidines CDP and UDP over purines ADP and GDP. This observation was first made for the E. coli RNR enzyme (**Zimanyi et al., 2016**), and we now find the same molecular mechanism of substrate specificity in play with the human enzyme.

CDP density is clearly observed in the active site (**Figure 5A**). As with other RNRs, substrate binding does not require $Mg^{2+}$ or other cations to counter balance the negative charge of the phosphates. Instead, the phosphates of CDP are bound through contacts with the backbone and/or side chains of Ser202, Thr604, Ala605, Ser606, and Thr607 (**Figure 5C**). Additionally, Arg293 on loop 2 (Arg298 in E. coli) reaches over the cytosine base to hydrogen bond with the phosphates, providing both a stacking interaction with the base and a positive-counter charge to the negatively charged substrate phosphates. The same interaction of Arg298 with the substrate phosphates is observed in E. coli for all four substrate-effector pairs (**Zimanyi et al., 2016**), leading to the proposal that

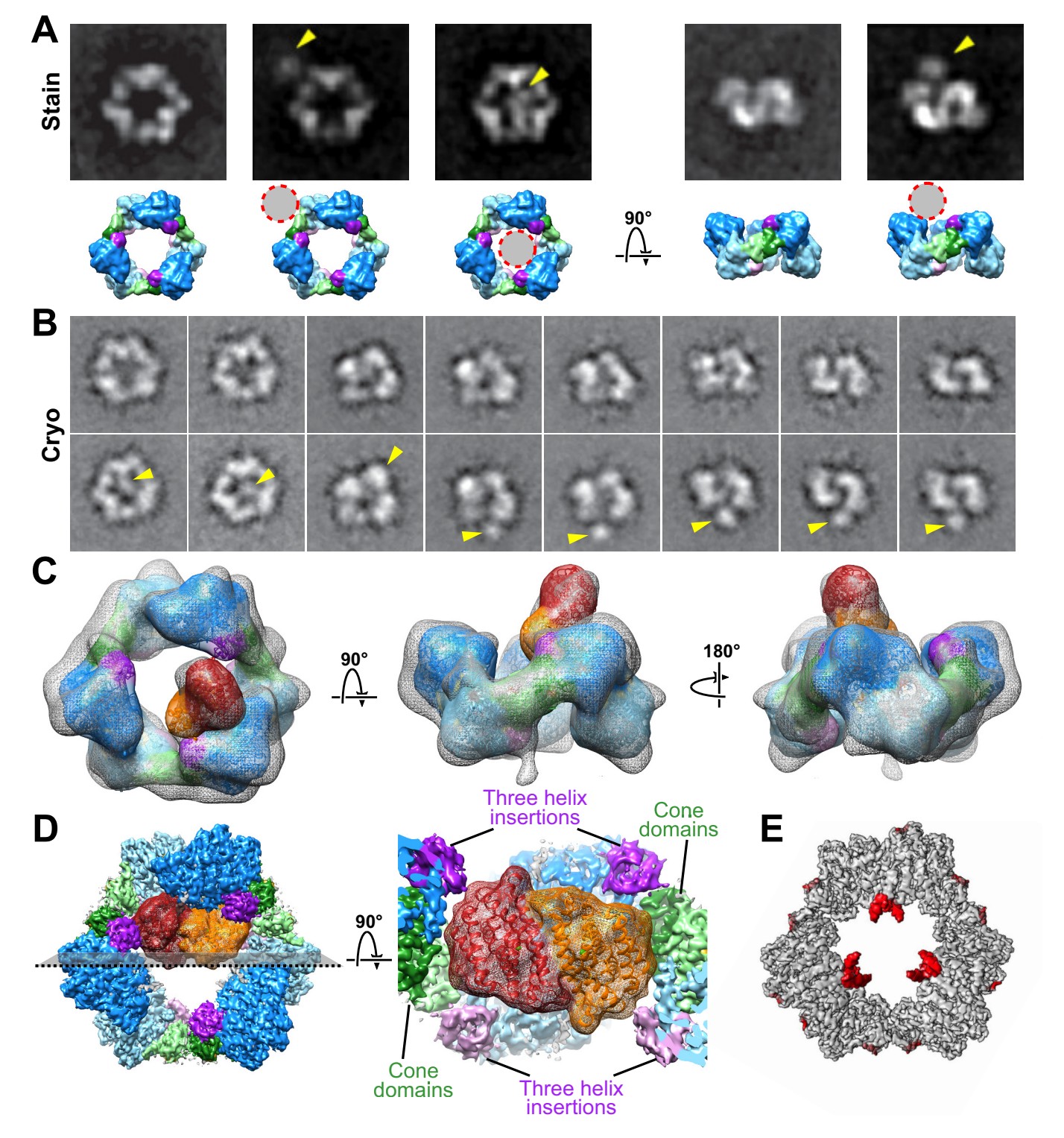

**Figure 4.** Interaction of β₂ with α₆ rings. (A) ISAC averages of negatively stained α with β in the presence of both 3 mM ATP and 0.05 mM dATP (same concentrations used for cryo-EM structure) reveal variability in both occupancy and location of β. Bottom row presents corresponding views of the α₆ ring as a low-resolution surface with circles highlighting the position of β₂ in the averages. (B) Similar views seen in ISAC averages from cryo-EM specimens of α with β and ClFTP. (C) 3D cryo-EM reconstruction of α with β and ClFTP. (D) Model of β₂ docked with an α₂ in the α₆ ring is tightly constrained by proximity of cone domains (green) and the three-helix insertion (purple) of α₂ and adjacent α subunits in the ring. (E) Variance map shows additional partially ordered density for C-terminal tails of α protruding into the central opening of the α₆ ring.

DOI: https://doi.org/10.7554/eLife.31502.013

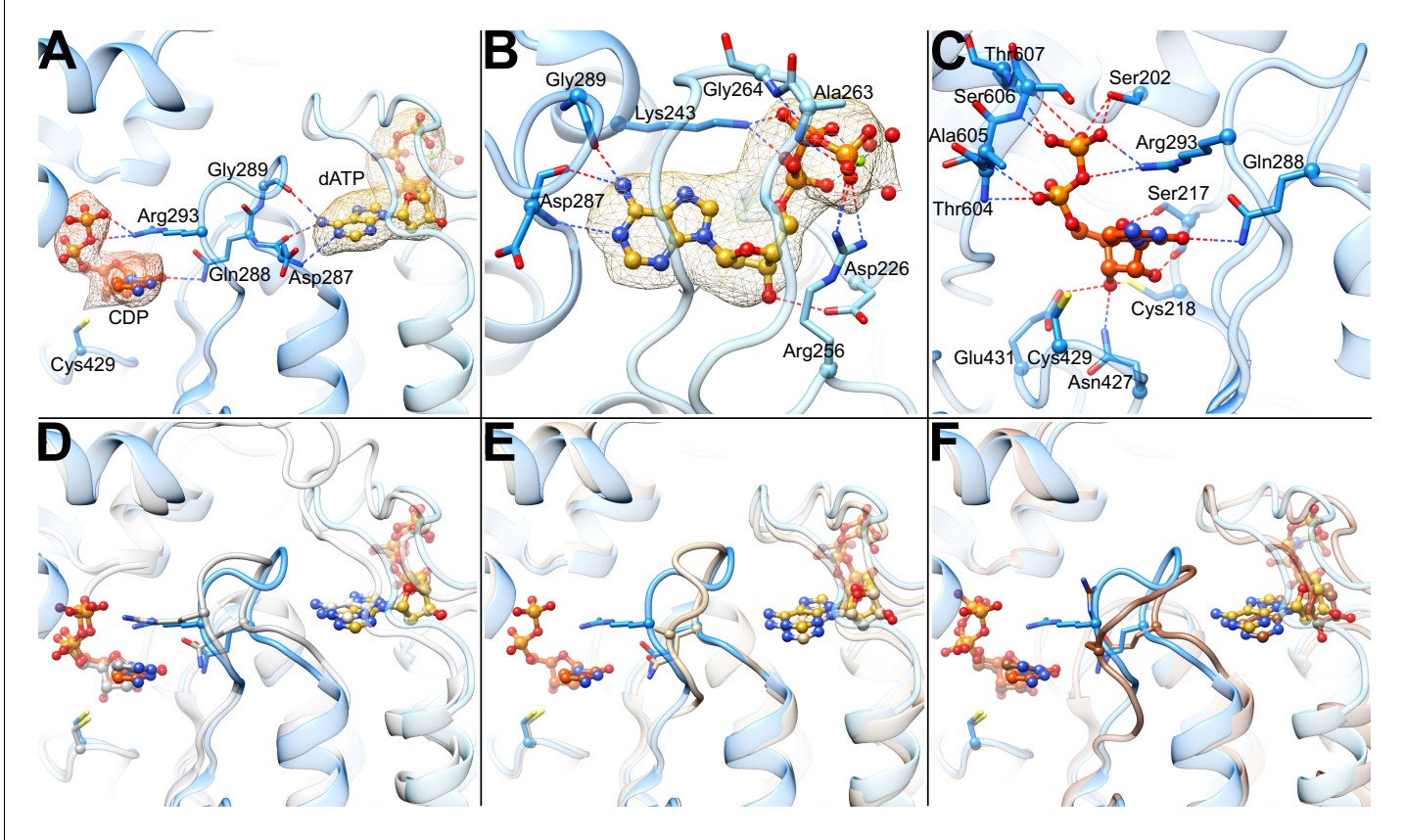

**Figure 5.** Determinants of substrate specificity are conserved from *E. coli* to human. (**A**) Residues of human α (blue) interacting with CDP (carbons in orange) in the active site and dATP (carbons in yellow) in the specificity site. Density for CDP in orange mesh and for dATP in yellow mesh. (**B**) Zoom in on dATP in the specificity site. Water molecules and oxygen atoms are in red, nitrogen in blue, magnesium in green, and phosphate in gold. (**C**) Zoom in on CDP in the active site. (**D**) Overlay of human α from the α$_6$ EM structure in blue with *E. coli* α from the α$_4$β$_4$-CDP-dATP cocrystal structure in gray (PDB: 5CNS) shows a nearly identical loop 2 conformation positioning Gln288 and Arg293 (Gln294 and Arg298 in *E. coli*). (**E**) Overlay of human α from the α$_6$ EM structure in blue with crystal structure of human α with N- and C-termini truncated (residues 77–742) cocrystallized with dATP in tan (PDB: 2WGH) shows similar positioning of dATP but an altered conformation of loop 2 in the absence of bound CDP. The CDP shown is from the α$_6$ EM structure. (**F**) Overlay of human α from the α$_6$ EM structure in blue with equivalent residues of yeast α structure with CDP and AMPPNP in brown (PDB: 2CVU) shows a conformation of loop 2 that is distinct from that seen in structures of *E. coli* and human α.

DOI: https://doi.org/10.7554/eLife.31502.014

Arg298 is a molecular latch that seals the active site for radical chemistry when the cognate substrate-effector pairs are bound. Consistent with this idea, mutation of Arg298 to Ala in *E. coli* abolishes activity for all four substrates (*Zimanyi et al., 2016*).

The substrate ribose modeled in the 3'-endo state is positioned by contacts with Glu431 (the catalytic acid/base) (*Lawrence et al., 1999*; *Licht and Stubbe, 1999*), Cys218 (one of the cysteines that forms a disulfide during catalysis), the side chain of Asn427, and the backbone carbonyl of Ser217 (*Figure 5C*). The Ser217 side chain also appears to contact the ribose O4', whereas a Ser rotamer faces away from the substrate in previously determined crystal structures of α from *E. coli*, yeast, and human (*Xu et al., 2006*; *Fairman et al., 2011*; *Zimanyi et al., 2016*). The Cys218-Cys444 disulfide appears to be reduced, and Cys429 (the thiyl radical forming cysteine) is ~4.3 Å from ribose C3', the site of hydrogen atom abstraction, which initiates catalysis (*Minnihan et al., 2013b*). Cys429 is also within hydrogen bonding distance Tyr737, which is adjacent to Tyr738. Both tyrosines are essential for RT (*Minnihan et al., 2013b*). As mentioned above, recognition of the cytosine base is mediated through Gln288 (Gln294 in *E. coli*).

## Discussion

The mechanisms of allosteric regulation of RNRs are both fascinating and complex. Substrate specificity regulation is particularly intriguing – how does the binding of dATP or ATP to the allosteric specificity site 15 Å away from the active site increase the preference of the active site for a pyrimidine substrate whereas TTP and dGTP binding promote purine substrates? Although the rules of regulation are conserved (*Figure 1B*) and the locations of the substrate and effector binding sites are conserved (*Figure 1C,D*), it has not been clear, even for prokaryotic versus eukaryotic class Ia RNRs, if the molecular mechanisms that afford the 'rules' will be the same. In part, the issue has been that obtaining structural data to visualize the molecular basis of allosteric specificity regulation has been nontrivial. To date, there are only a handful of structures of RNRs that have clear density for the loop responsible for the allosteric communication (loop 2) (*Xu et al., 2006*; *Zimanyi et al., 2016*), which has led to conflicting mechanistic proposals (*Xu et al., 2006*; *Ahmad et al., 2012*; *Zimanyi et al., 2016*). The near-atomic resolution structure presented here provides the first visualization of RNR structure from a eukaryotic RNR that has a well-ordered loop 2. Using our structure, which has CDP

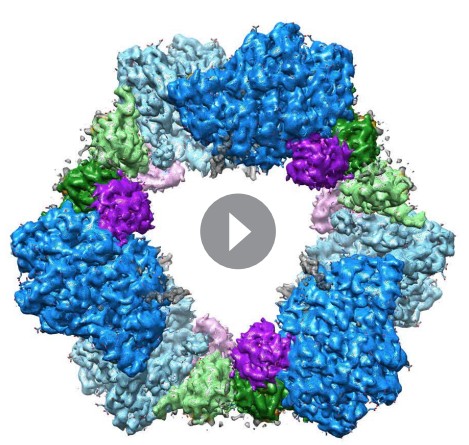

**Video 2.** Overview of the cryo-EM density and detailed view of the bound CDP substrate, dATP specificity effector, and loop 2 conformation. Starting from a view down the three-fold symmetry axis, the structure rotates while zooming through the center of the ring for a view of the loop 2 region that confers substrate preference as seen in *Figure 5A*, and then rocks back and forth over a range of 30°. Density is shown as mesh colored according to regions of the structural model as in *Figure 1*.
DOI: https://doi.org/10.7554/eLife.31502.015

bound in the active site, we can investigate whether the molecular mechanism for pyrimidine substrate preference is conserved between human RNR and the well-studied *E. coli* enzyme, resolving a controversy as well as providing key insight into this captivating form of allosteric regulation. The molecular basis for allosteric regulation of activity has been similarly enigmatic. Prior to this study, we knew that human RNR forms an $\alpha_6$ state in the presence of allosteric activity effectors ATP and dATP, but it was not clear why the stable dATP-induced $\alpha_6$ state was inactive. Here our 3.3-Å resolution cryo-EM structure of the human RNR enzyme in a dATP-inhibited $\alpha_6$ state has considerably surpassed the resolution of a previous 9-Å resolution crystal structure (*Ando et al., 2016*) and has allowed us to interrogate the molecular basis of dATP-associated inactivity for human RNR. We also use these data, along with lower resolution EM structures of human RNR $\alpha_6$ rings in the presence of the anticancer drug CIFTP and the radical storage $\beta_2$ subunit, to consider the implications of these findings for anticancer therapies.

Our near-atomic resolution structure of human $\alpha_6$ is the first structure of the human enzyme with the CDP/dATP substrate/effector pair bound. As mentioned above, it also represents one of the few RNR structures with a well-ordered loop 2, the region of the structure responsible for communication between the specificity effector and its cognate substrate. The conformation of loop 2 observed between the base of the dATP specificity effector and the base of the CDP substrate indicates that communication of CDP binding preference from the specificity effector-binding site is conserved with *E. coli* $\alpha_2$ (*Figure 5D*). In both systems, backbone atoms of this loop 'read out' the adenine base and position Gln288 into the active site to recognize the cytosine of CDP. With the CDP appropriately bound, Arg293 seals the active site for radical chemistry. A previously solved structure of human $\alpha_2$ with dATP bound in the specificity site, but without substrate (*Fairman et al., 2011*), shows Gln288 positioned for pyrimidine recognition (CDP or UDP), but in the absence of substrate, Arg293 is disordered and the C-terminal portion of loop 2 has not yet adopted the conformation that would fully stabilize the substrate-effector pair (*Figure 5E*). Importantly, a crystal structure of yeast $\alpha_2$ with ATP analog AMPPNP and CDP bound exhibits a conformation unlike that seen in the human or *E. coli* structures (*Figure 5F*). Substantial differences between yeast and *E. coli* substrate-

effector bound RNR structures have been previously reported (*Xu et al., 2006*; *Ahmad et al., 2012*; *Zimanyi et al., 2016*) and opposite roles for the conserved Gln of loop 2 have been proposed. For *E. coli*, it is believed that the positioning of the Gln side chain into the active site promotes CDP/UDP binding through hydrogen bonding while hindering ADP/GDP binding due to steric bulk. However, for yeast, it has been proposed the Gln side chain points into the active site to increase (not decrease) the affinity of ADP. Thus, one goal of this work was to evaluate whether eukaryotic RNRs, such as human and yeast, use Gln and loop 2 differently from their prokaryotic counterparts. In other words, we wished to establish the conformation of loop 2 and Gln for the CDP-dATP substrate-effector pair in the human enzyme and compare it to yeast and *E. coli*. As described above, we find that the human RNR communicates CDP preference in an identical fashion as *E. coli*. Therefore, whether or not the yeast enzyme does employ Gln differently, it is not the case that all eukaryotic RNRs will be different from prokaryotic RNRs. Notably, with the addition of this CDP-dATP bound structure, there are now six RNR structures for which loop 2 is well ordered. Four are from the *E. coli* class Ia RNR (CDP-dATP, UDP-dATP, GDP-TTP, and ADP-dGTP, listed as substrate-specificity effector pair) (*Zimanyi et al., 2016*), one from *Thermatoga maritima* class II RNR (GDP-TTP) (*Larsson et al., 2004*), and this one for human class Ia RNR (CDP-dATP). It was previously reported that the two GDP-TTP structures are highly similar despite being from different RNR species and classes. Taken together with these results, we postulate that the molecular mechanism of allosteric specificity regulation will be conserved among class I and II RNRs; that structural differences observed to date are more likely due to crystal disorder/lattice contacts rather than real variations in mechanism. Single particle cryo-EM, a technique for which lattice contacts are not an issue, thus provides valuable insight about how an unrestrained regulatory feature, such as loop 2, responds to substrate/effector binding. Moving forward, we suspect that cryo-EM will be increasingly used to investigate allosteric regulation.

Whereas our results support a conserved mechanism for allosteric regulation of specificity among class Ia RNRs from different species, they argue against conservation of the mechanism for allosteric activity regulation. For the human enzyme in the presence of dATP, we see no evidence of *E. coli*-like $\alpha_4\beta_4$ structures (*Figure 2A*) nor do we see anything that resembles the $\alpha_4$ oligomeric state that was recently reported for dATP-inhibited *Pseudomonas aeruginosa* RNR (*Johansson et al., 2016*). Instead, we observe $\alpha_6$ and some $\alpha_6\beta_2$. In agreement with quantitative studies of human RNR by SAXS (*Ando et al., 2016*), EM analysis shows that $\alpha_6$ rings form in the presence of both dATP and ATP, or ClFTP, and that addition of $\beta_2$ does not disrupt these stable rings. Additionally, our EM data show that $\beta_2$ can sit in multiple positions around the outside of the $\alpha_6$ ring, but appears unable to penetrate into the inside of the ring structure where it could initiate chemistry on $\alpha_2$. Thus, instead of 'holding $\beta_2$ at arm's length' to prevent RT in the presence of dATP as in *E. coli* RNR, the human enzyme holds $\alpha_2$ subunits together in a circle in the presence of dATP to exclude $\beta_2$ from accessing $\alpha_2$ (*Figure 2*).

The cryo-EM structure furthermore offers an explanation for $\beta_2$ exclusion from the inside of the $\alpha_6$ ring. We find that $\beta_2$ access is impaired by the 60-Å constriction at the surface of the $\alpha_6$ ring and by the disordered C-terminal tails of $\alpha$ that extend into the ring. Also, the ability of $\beta_2$ to assume a catalytically relevant position with respect to $\alpha$ is restricted by the cone domains at the $\alpha_2$-$\alpha_2$ interface and the three-helix insertion motif of adjacent $\alpha_2$ subunits. In short, when the ring is stable and cannot expand, $\beta_2$ cannot fit and assume a catalytically relevant position. Thus, the weak association between the cone domains in the presence of ATP should allow $\beta_2$ to further destabilize the $\alpha_6$ ring by inserting itself into the central opening to form an active complex with $\alpha_2$. In contrast, tight association of the cone domains in the presence of dATP would preclude productive $\beta_2$ interaction with $\alpha_2$ (*Figure 2B*).

These structural results also offer insight into inhibition of RNR by the diphosphate and triphosphate forms of the therapeutics clofarabine (ClFDP and ClFTP), cladribine (ClADP and ClATP), and fludarabine (FlUTP) that are known to impart $\alpha$ hexamers with enhanced stability (*Aye and Stubbe, 2011*; *Aye et al., 2012*; *Wisitpitthaya et al., 2016*). Given that the well-studied RNR inhibitors gemcitabine-diphosphate and hydroxyurea are known to target the active site on $\alpha$ and the tyrosyl radical on $\beta_2$, respectively (*van der Donk et al., 1998*; *Offenbacher et al., 2014*), the possibility that some RNR inhibitors might act by targeting the allosteric activity site on $\alpha$ was intriguing to investigate. ClF and ClA are cytotoxic in a cell line expressing wild-type $\alpha$, whereas a cell line expressing an $\alpha$ mutation in its allosteric activity site (Asp57Asn) is resistant to the drugs (*Wisitpitthaya et al.,*

*2016*), consistent with the allosteric activity site being targeted by these molecules. From our data, we can now rationalize that molecules that stabilize the hexamer by binding to the cone domain will inhibit human RNR by impairing the ability of $\beta_2$ to access $\alpha$. Thus, designing inhibitors to bind to the allosteric site in the cone domain appears to represent a new and attractive method for inhibiting human RNR in vivo.

In summary, advances of cryo-EM methodology have allowed us to obtain a near-atomic resolution structure that has provided remarkable insight into the allosteric regulation of the human RNR enzyme. Although detailed structural information about the active $\alpha$-$\beta$ RNR complexes is still an important missing piece to the puzzle, our understanding of RNR inactive states has advanced substantially with views of beautiful yet distinct ring structures.

## Materials and methods

### Materials

5-$^3$H]-CDP was obtained from ViTrax. hTrx1 and hTrxR1 were isolated as previously described (*Ando et al., 2016*).

### Methods

#### RNR expression, purification, and activity assays

His$_6$-tagged forms of human $\alpha$ and $\beta$ were expressed from plasmid pET-28a (*Wang et al., 2007*) in *E. coli* strain BL21(DE3)-RIL (Stratagene) and purified by Co$^{2+}$-affinity chromatography as previously described (*Aye and Stubbe, 2011*; *Ando et al., 2016*). $\alpha$ typically has a specific activity of 600 ~ 850 nmol dCDP/min/mg and reconstituted $\beta$ has a specific activity of 2000 ~ 2400 nmol dCDP/min/mg (*Figure 3—figure supplement 2*). $\alpha$ is stored in 50 mM Tris, pH 7.6, 15 mM MgCl$_2$, 100 mM KCl, 5 mM DTT, 5% glycerol at 8.4 mg/mL (92 µM) for all cryo-EM experiments. Activities of $\alpha$ were determined by measuring the reduction of [5-$^3$H]-CDP with a 5-fold molar excess of $\beta$. To mimic the enzyme concentrations used in the cryo-EM studies, the reaction mixture at 37°C of 180 µL contained 14 µM $\alpha$, 70 µM $\beta$, 5 mM [5-$^3$H]-CDP (3633 cpm/nmol). ATP and/or dATP at the different concentrations, 25 µM human thioredoxin (hTrx1), 0.2 µM thioredoxin reductase (hTrxR1), 2 mM NADPH in 50 mM HEPES, pH 7.6, 15 mM MgCl$_2$, 150 mM KCl. The mixtures were incubated for 30 s at 37°C after addition of $\beta$, and the reaction was initiated by addition of [5-$^3$H]-CDP. Aliquots (40 µL) were taken at 0, 1, 2, and 3 min and quenched in a boiling water bath for 2 min. dC formation was analyzed by the method of (*Steeper and Steuart, 1970*) and quantified by scintillation counting.

#### Negative stain EM studies with human $\alpha$ in absence of $\beta_2$

Purified recombinant $\alpha$ at 6 µM (or 10 µM $\alpha$ in the case of the ClFDP/dGTP experiment) was incubated at 37°C for 2 min with effector nucleotides at the indicated concentrations (*Figure 3—figure supplement 1*) in 50 mM HEPES, pH 7.6, 15 mM MgSO$_4$, 1 mM EDTA, 5 mM DTT and then further diluted to ~17 µg/mL (0.19 µM) in buffer with nucleotides. Preparation of specimens with ClFDP/dGTP and with 1–10 mM ATP were previously described in (*Aye et al., 2012*) and (*Ando et al., 2016*). 5 µL of the diluted mixture was applied to carbon-coated 300 mesh Cu/Rh grids (Ted Pella) that had been glow discharged immediately before use. After allowing ~1 min for protein adsorption, the solution was blotted, washed three times with 5 µL 2% uranyl acetate (Ted Pella), and incubated for 1 min in the final uranyl acetate wash before applying a second carbon layer, blotting, and air drying (*Tischendorf et al., 1974*). Images were acquired of each specimen at a magnification of 50,000× on a Tecnai F20 Twin (FEI) operated at 120 kV equipped with a US4000 CCD detector (Gatan). Power spectra were examined for drift and astigmatism and those with defocus values of approximately 0.6 µm as estimated by SPIDER (*Frank et al., 1996*) or ACE2 (*Mallick et al., 2005*) were further processed. Particles were selected, windowed, downsampled by a factor of 3 to a pixel size of 6.51 Å (the smaller TTP particles were downsampled by 2 to 4.34), and normalized with EMAN2 (*Tang et al., 2007*) and SPARX (*Hohn et al., 2007*). After initial reference-free alignment, K-means classification, and particle cleaning with SPARX, ISAC (*Yang et al., 2012*) was used to generate final class averages. Poorly resolved ISAC averages were eliminated, and the remaining averages were aligned and sorted for visual comparison (*Figure 3—figure supplement 1*).

## Negative stain EM studies of the dATP-inhibited human $\alpha_6$ ring with $\beta_2$

Negative stain EM studies (*Figure 4A*) were conducted for $\alpha$ in the presence of $\beta$ using the same inhibitory concentrations of dATP and ATP that were used for cryo-EM. Following 1 min incubation at 37°C, 3 µM $\alpha$ was combined with a pre-warmed mixture of buffer (50 mM HEPES, pH 7.6, 15 mM MgCl$_2$, 50 mM KCl, 5 mM DTT) and final concentrations of 3 mM ATP, 0.05 mM dATP, and 1 mM CDP, followed by addition of 3 µM $\beta$. The mixture was incubated at 37°C for 2 min followed by further ~20 fold dilution in buffer containing nucleotides at the same concentrations before applying 5 µL to a continuous carbon coated grid and staining as described above. Stained specimens were imaged using a Tecnai T12 Twin electron microscope (FEI) equipped with a LaB$_6$ filament, operated at 120 kV acceleration voltage and 52,000× magnification. Images were recorded on a TemCam-F416 CMOS detector (TVIPS). Particle images were selected using DoGPicker (*Voss et al., 2009*) and analyzed with SPARX. ISAC averages corresponding to views from above the ring (along the three-fold symmetry axis) and from an orthogonal direction (along the two-fold axis) provided information about the presence and position of $\beta_2$ density (*Figure 4A*).

## Cryo-EM studies of human $\alpha_6$ with ATP, dATP, and CDP

### Specimen preservation

To prepare cryo-EM specimens, $\alpha$ and a mix of ATP, dATP, and CDP were separately incubated at 37°C for 1 min and then combined to a final composition of 14 µM $\alpha$, 0.05 mM dATP, 3 mM ATP, 1 mM CDP in 50 mM HEPES, pH 7.6, 15 mM MgCl$_2$, 1 mM EDTA, 5 mM DTT, and 50 mM KCl. The addition of KCl to the buffer prevented clumping of particles. The mixture was then incubated for an additional 2 min at 37°C. In a cold room at >90% relative humidity, 2.4 µL of the protein-nucleotide mixture was applied to a CFlat 400 mesh Cu grid supporting a carbon film of 2 µm holes with 2 µm spacing (Protochips) (*Quispe et al., 2007*) that had been plasma cleaned in a Solarus 950 (Gatan) at 25 W for 10 s in 75% Ar, 25% O$_2$ gas mixture and then immediately before use was glow discharged (*Dubochet et al., 1971*) at 20 mA for 30 s in an EMITech K100X. The grid was manually blotted with filter paper and plunged into liquid ethane (*Adrian et al., 1984*).

### Imaging

Cryo-EM specimens of $\alpha$ with ATP, dATP and CDP were imaged at 22,500× magnification (resulting in a pixel size of 1.31 Å on the specimen scale) with underfocus values between 0.8 and 2.8 µm, using a Titan Krios electron microscope (FEI) operating at an accelerating voltage of 300 kV (*Table 1*). Automated data collection was carried out with Leginon (*Suloway et al., 2005*) and a total of 2144 micrographs were recorded on a K2-Summit direct electron detector (Gatan) operated in counting mode. A total accumulated dose of 44 electrons per Å$^2$ was fractionated into 38 frames over a 7.6 s exposure time.

### Frame alignment and defocus estimation

Full frames of each dose-fractionated exposure were aligned and summed with motion_corr (*Li et al., 2013*) (*Table 1*). Defocus of aligned frame sums was estimated with CTFFIND3 (*Mindell and Grigorieff, 2003*) and CTER (*Penczek et al., 2014*).

### Particle selection

An initial set of 5,000–10,000 particles were selected manually from a subset of frame sums and used to calculate 2D class averages with ISAC (*Yang et al., 2012*). These averages were used as templates for automated particle selection using FindEM (*Roseman, 2003*) in Appion (*Lander et al., 2009*). Initial screening of particles was performed using multivariate statistical analysis image clustering. A second round of image screening was performed using ISAC on 4-fold decimated images. ISAC class averages lacking clear features, not resembling possible projections of a macromolecular complex (e.g., ice contamination), or showing density corresponding to more than one particle were eliminated. The 43,885 images included in the remaining ISAC averages were used for further image processing in the SPARX and RELION EM image processing packages (*Hohn et al., 2007*; *Scheres, 2012*) (*Table 1*). These screened images were processed with the movie refinement and particle polishing routines in RELION and 'shiny' images comprising all but the first four detector frames were used for all further image-processing steps.

## Map refinement

A cryo-EM map of the dATP-inhibited $\alpha_6$ ring with a final resolution of 3.3 Å (at FSC = 0.143) was calculated using SPARX following a protocol comprising two parameter refinement stages (*Figure 3A* and *Figure 3—figure supplement 3*). In the first stage, the cryo-EM image data were divided into two non-overlapping subsets and orientation parameters for each subset were independently refined starting from an ~25 Å initial reference, which was calculated *ab initio* through stochastic hill climbing refinement (*Elmlund et al., 2013*) starting from a set of ~10,000 images to which random Euler angle values were assigned, thereby eliminating any possibility of reference bias. At each iteration, low-pass filtering for the partial volumes was set to the frequency corresponding to the FSC = 0.5 value between two quasi-independent partial maps. This initial stage was followed by local orientation parameter optimization with template volume amplitude adjustment. Amplitude correction factors were determined using the rotationally averaged power spectrum computed for an initial pseudoatomic $\alpha_6$ model generated by fitting crystal structure of human $\alpha$ (PDB: 3HNF chain B) into a lower resolution EM map. Local resolution was computed using the local resolution calculation function implemented in SPARX (*Figure 3—figure supplement 4*).

## Model refinement

Coordinates from the crystal structure of human $\alpha$ (PDB: 3HNF) were docked into the EM reconstruction with UCSF Chimera (*Goddard et al., 2007*). The initial model was adjusted by hand using COOT (*Emsley et al., 2010*) and further adjusted through refinement in reciprocal space with phenix.refine (*Adams et al., 2010*). To speed up refinement in reciprocal space, the map was trimmed using e2proc3d.py (*Tang et al., 2007*) to 160 × 160 × 96 voxels, dimensions slightly larger than the oblate shape of the ring. dATP was modeled in the activity and specificity sites. Although at 3.3-Å resolution ATP and dATP are difficult to distinguish, dATP appeared to be the better fit and is expected to have a higher affinity for both sites (*Brown and Reichard, 1969b*; *Reichard et al., 2000*; *Kashlan and Cooperman, 2003*). CDP was modeled in the active site. Final rounds of model building and refinement were done with COOT and phenix.real_space_refine version 1.11.1-dev-2650 (*Adams et al., 2010*). In real space refinement, resolution was set to 3.0 Å, electron scattering table was selected, NCS constraints for the six $\alpha$ subunits were automatically detected and refined, secondary structure hydrogen bonds were relaxed to 0.4 sigma, and secondary structure restraints were manually defined by comparison with existing crystal structures of human, yeast, and *E. coli* $\alpha$ and with secondary structure restraints that were determined automatically by PHENIX. Definitions for CDP and dATP from the CCP4 monomers library were modified to restrain ideal phosphate dihedral angles and specify alternate conformations for sugar pucker (C2′-endo/C3′-exo and C2′-exo/C3′-endo). Phosphates of dATP were modeled with an octahedrally coordinated $Mg^{2+}$ ion to three water molecules, restrained to ideal distances (2.1 Å) and angles (90°). Model quality was evaluated using Molprobity (*Chen et al., 2010*), CaBLAM (*Richardson and Richardson, 2013*), and EMRinger (*Barad et al., 2015*) (*Table 1*). The final model is a good fit to the map (*Figure 3A*) and contains two residues of tag, residues 1 through 743 of the 792 residues of human $\alpha$, twelve dATP molecules, and six CDP molecules (*Table 1*). Figures of the model and map were rendered with UCSF Chimera.

## Preparation of cryo-EM specimens, imaging, and analysis of the ClFTP-inhibited $\alpha_6$ ring with $\beta_2$

### Specimen preservation

15 μM $\alpha$ and 16.5 μM $\beta$ were added to a solution containing 100 μM dGTP in 50 mM HEPES, pH 7.6, 15 mM $Mg_2SO_4$, 1 mM EDTA, 5 mM DTT and incubated at 37°C for 2 min before addition of a 5-fold molar excess of ClFTP (synthesized from ClF (AK Scientific) as described in [*Aye and Stubbe, 2011*]) over $\alpha$. This mixture was further incubated at 37°C for 2 min before dilution 75-fold into 50 mM HEPES, pH 7.6, 15 mM $Mg_2SO_4$, 150 mM NaCl containing 0.5 mM ATP to mimic the elution buffer used in gel filtration experiments of $\alpha$ and $\beta$ (*Wang et al., 2007*; *Aye and Stubbe, 2011*). 3.2 μL of the mixture was applied to a freshly glow discharged thin continuous carbon film supported by a 400 mesh CFlat grid with 2 μm holes (Protochips), blotted manually, and plunged in liquid ethane.

### Imaging

Images were acquired on a Tecnai F20 Twin (FEI) set up as above for negative stain acquisition, with 103 micrographs acquired manually with exposures of ~20 e⁻/A² on Kodak SO163 film and digitized at 1.27 Å/pixel on a Nikon CoolScan 9000ED.

### Analysis

Particles were selected, windowed, downsampled to 5.08 Å/pixel, and normalized with EMAN2. Reference-free 2D alignment and K-means clustering was used for initial image screening, and the remaining 29,027 particles were classified with ISAC (*Yang et al., 2012*). ISAC averages that represent views with and without additional $\beta_2$ density were extracted (*Figure 4B*). ISAC groups that clearly had $\beta_2$ density were subjected to 3D refinement with SPARX (*Hohn et al., 2007*) using a low-pass filtered initial map of $\alpha_6\beta_2$ generated with an identically prepared negative stain specimen of $\alpha$ and $\beta$ with ClFTP using the Random Conical Tilt procedure (*Radermacher et al., 1987*). In USCF Chimera, our dATP-inhibited $\alpha_6$ model was docked into the density map, and the human $\beta_2$ model (PBD: 2UW2) was positioned into the poorly resolved additional density that likely results from variable $\beta_2$ position with respect to the $\alpha_6$ ring (*Figure 4C*).

## Preparation of cryo-EM specimens, imaging, and analysis of human $\alpha$ with dATP

### Specimen preservation

Cryo-EM specimens of $\alpha$ with dATP (*Figure 3—figure supplement 1D*) were prepared as follows. 20 $\mu$M $\alpha$ was diluted in 50 mM HEPES, pH 7.6, 15 mM MgCl$_2$, 5 mM DTT, and 0.1 mM dATP. 5 $\mu$L of the mixture was applied to a Quantifoil grid that had been washed first with 5 $\mu$L of the dATP buffer mixture. The grid was blotted for 3 s in a Vitrobot (FEI) at 25°C, 100% humidity, and then plunged in liquid ethane.

### Imaging

Using EPU software, 860 images were acquired on a Titan Krios (FEI) operated at 300 kV and a magnification of 29,000× with a backthinned Falcon I detector (FEI) with 1 s exposures of 21 e⁻/A² at 2.30 Å/pixel.

### Analysis

Defocus of images was estimated with CTER (*Penczek et al., 2014*). Particles were identified automatically with PARTICLE using three distinct reference projections calculated from the yeast $\alpha_6$ crystal structure (PDB: 3PAW) filtered to 30 Å resolution (*Fairman et al., 2011*). Particle images were windowed, phase-flipped, downsampled to 4.6 Å/pixel, and normalized with EMAN2. Reference-free 2D alignment, K-means clustering, and ISAC were used for initial image screening, and the remaining 58,727 particles were classified with a final round of ISAC (*Yang et al., 2012*). 2D averages indicate the presence of conformational heterogeneity (*Figure 3—figure supplement 1D*) consistent with what was seen for this nucleotide condition in negative stain (*Figure 3—figure supplement 1C*).

## Acknowledgements

We thank N Ando for helpful dialog about conditions for specimen preparation, M Funk for assistance preparing negative stain specimens, and S Thompson for assistance selecting particles from the ClFTP cryo-EM specimen. Imaging of the dATP cryo-EM specimen was performed at Janelia Research Campus with assistance from Z Yu and J Cruz. All other imaging was performed at The Scripps Research Institute cryo-EM facility with support from National Institutes of Health (NIH) through the National Center for Research Resources P41 program (RR017573). Micrograph negatives were digitized at Brandeis University with access generously provided by N Grigorieff. This research was supported by NIH grants GM29595 (to JS) and GM67167 (to FJA). CLD is an HHMI investigator.

## Additional information

### Funding

| Funder | Grant reference number | Author |
| --- | --- | --- |
| Howard Hughes Medical Institute | Investigator | Catherine L Drennan |
| National Institutes of Health | Research Project (R01), GM67167 | Francisco Asturias |
| National Institutes of Health | Research Project (R01), GM29595 | JoAnne Stubbe |

The funders had no role in study design, data collection and interpretation, or the decision to submit the work for publication.

### Author contributions

Edward J Brignole, Conceptualization, Resources, Formal analysis, Validation, Investigation, Visualization, Methodology, Writing—original draft, Writing—review and editing; Kuang-Lei Tsai, Formal analysis, Validation, Investigation, Visualization, Writing—review and editing; Johnathan Chittuluru, Pawel A Penczek, Software, Formal analysis, Writing—review and editing; Haoran Li, Resources, Validation, Investigation, Writing—review and editing; Yimon Aye, Resources, Writing—review and editing; JoAnne Stubbe, Conceptualization, Resources, Supervision, Funding acquisition, Validation, Project administration, Writing—review and editing; Catherine L Drennan, Conceptualization, Formal analysis, Supervision, Funding acquisition, Validation, Visualization, Methodology, Writing—original draft, Project administration, Writing—review and editing; Francisco Asturias, Conceptualization, Software, Formal analysis, Supervision, Funding acquisition, Validation, Investigation, Visualization, Methodology, Project administration, Writing—review and editing

### Author ORCIDs

Edward J Brignole (iD) http://orcid.org/0000-0002-4285-6128
Catherine L Drennan (iD) http://orcid.org/0000-0001-5486-2755

### Decision letter and Author response

Decision letter https://doi.org/10.7554/eLife.31502.022
Author response https://doi.org/10.7554/eLife.31502.023

# Additional files

### Supplementary files

• Transparent reporting form
DOI: https://doi.org/10.7554/eLife.31502.016

### Major datasets

The following datasets were generated:

| Author(s) | Year | Dataset title | Dataset URL | Database, license, and accessibility information |
| --- | --- | --- | --- | --- |
| Edward J Brignole, Catherine L Drennan, Francisco Asturias, Kuang-Lei Tsai, Pawel A Penczek | 2018 | Human ribonucleotide reductase large subunit (alpha) with dATP and CDP | http://www.ebi.ac.uk/pdbe/entry/emdb/EMD-7006 | Publicly available at the Electron Microscopy Data Bank (accession no. EMD-7006) |

| Edward J Brignole, Catherine L Drennan, Francisco Asturias, Kuang-Lei Tsai, Pawel A Penczek | 2018 | Human ribonucleotide reductase large subunit (alpha) with dATP and CDP | http://www.rcsb.org/pdb/search/structid-Search.do?structureId=6AUI | Publicly available at the RCSB Protein Data Bank (accession no. PDB 6AUI) |
|---|---|---|---|---|

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
