## [Decision Letter]

Thank you for submitting your article "3.3-Å resolution cryo-EM structure of human ribonucleotide reductase with substrate and allosteric regulators bound" for consideration by *eLife*. Your article has been reviewed by two peer reviewers (one of whom was Werner Kühlbrandt), and the evaluation has been overseen by John Kuriyan as the Senior Editor and the Reviewing Editor.

The Reviewing Editor has drafted this decision to help you prepare a revised submission.

Summary:

This paper presents the 3.3 Å cryo-EM structure of human ribonucleotide reductase (RNR) in dATP, CDP bound forms as well as, in a lower resolution structure, with the inhibitor CIFTP, as well as with the β subunit. This is an important study, because of the fundamental importance of RNR in DNA metabolism, and the fact that human RNR is a potential anti-cancer target. In addition, the enzyme is regulated by complex allosteric control, with two kinds of allosteric sites, one governing specificity and one governing activity. The allosteric control involves switching between different quaternary assemblies, and some aspects of this mechanism are different between the *E. coli* enzyme (particularly well studied) and the human enzyme (less well studied). X-ray structures are available for the *E. coli* homolog and for dimeric forms of the α and β subunits of human RNR, but so far there is no high-resolution structure of the hexameric human enzyme, which is relevant for allosteric regulation. At an estimated molecular mass of around 500 kDa, the hexamer of human RNR is an ideal target for structure determination by cryo-EM, and the present work substantially improves upon the low resolution reconstructions available previously. The structures confirm proposed models for the hexamer organization and reveal more resolved information of the cone domain interactions in the hexamer. The structure also confirms suggested models for how substrate should bind in the active site of mammalian enzymes.

The EM map appears to be of a very high quality. The final resolution of 3.3 Å is excellent, considering the pixel size (1.3 Å) and number of particles used (44,000 out of 150,000), and bearing in mind that cryo-EM maps are clearer and better than x-ray maps at the same nominal resolution. This is due to the quality of structure factor phases, which are determined directly.

Major concerns to address:

1) Given the power of cryoEM to resolve this complex with various substrates and effectors bound, it is surprising that the authors describe only one substrate/effector pair. The paper would be strengthened if the results were expanded to a complete set of cognate effector substrate pairs. If this has not been done, the revised manuscript should describe why this is the case. Note that we are not asking for the initiation of new experimental work at this stage, but if additional data are available then they would strengthen the paper. In any case, discussion of this point should be provided in the revised manuscript.

2) In the paper, as written, it is difficult for a non-specialist reader to discern what specifically is the advance reported in the paper, given the lower-resolution reconstruction of the hexameric α-subunit structure reported previously. There is some discussion of this in the concluding section of the paper, but the paper would be strengthened by bringing to the front a discussion of what is not known, and what conceptual advances are to be expected of the present work, beyond improving the resolution.

Please note the other important points to address, shown below. Many of these points concern the clarity of the manuscript and the accessibility to a non-specialist reader.

1) The paper is written in such a way that it is quite difficult for a non-specialist to appreciate the complex allosteric regulation of this enzyme, and to put the work in the context of previous results. The two kinds of allosteric sites (which are referred to throughout the paper) should be clearly explained, and the expectations as to which of these sites would be occupied (or not) in the hexameric structure should be discussed. The impact of the figures would be improved by accompanying schematic diagrams.

2) A schematic diagram should be provided that shows the allosteric modulators, and how they alter oligomeric states. Figure 1 shows specificity and activity effectors without explaining what they are.

3) Figure 1 shows a complicated feedback network in which different nucleotides affect the activity (and specificity?) of the enzyme. With the improved description of the enzyme, this diagram should be related to the known structural features (see the major concern noted above, which asks why the information on different effectors and substrates is limited).

4) In Figure 1, an equilibrium between different oligomeric states is shown, but without explanation of how and why ATP shifts this equilibrium.

5) Why was a mixture of dATP and ATP used to prepare the cryoEM sample? There is one CDP molecule per subunit, but it is not described where this came from. The logic of the sample preparation should be explained.

6) It would be interesting to know by how much the map resolution improved when 6-fold averaging was applied.

7) Additional figures showing densities of residues in important regions should be provided, such as the cone interactions and dATP interaction, as well as the loop 2 conformation and substrate coordination.

---

## [Author Response]

Major concerns to address:1) Given the power of cryoEM to resolve this complex with various substrates and effectors bound, it is surprising that the authors describe only one substrate/effector pair. The paper would be strengthened if the results were expanded to a complete set of cognate effector substrate pairs. If this has not been done, the revised manuscript should describe why this is the case. Note that we are not asking for the initiation of new experimental work at this stage, but if additional data are available then they would strengthen the paper. In any case, discussion of this point should be provided in the revised manuscript.

We agree that structures with other substrate-effector pairs will be important to complete the picture of specificity regulation in human RNR. However, those experiments, if possible by this approach, will require more complicated mixtures of nucleotides and we have not worked out the appropriate conditions for those structural studies.

To explain more fully, the dATP/ATP/CDP combination that we used in this study was empirically determined over the course of many years to provide stabilized, rigid alpha6 rings. dATP/ATP are unique among allosteric effectors of RNR in their ability to form alpha6 rings while simultaneously serving as the specificity effector for the substrate CDP. The other nucleotide effectors, TTP or dGTP, that specify GDP or ADP substrate preferences, do not stabilize the large, symmetric alpha6 rings (see Figure 2—figure supplement 1 and Ando et al., 2017), preferentially form alpha2 (150 kDa), and would make resolving loop2 and bound nucleotides by cryo-EM a substantial challenge. To obtain structures with TTP/GDP or dGTP/ADP substrate/effector pairs, in the context of an alpha6 ring, could require a carefully tuned combination with dATP and/or ATP effector at concentrations and where TTP or dGTP effectors are not displaced at the specificity site. Defining conditions that would allow us to achieve structures of these states would benefit from a deeper understanding of binding affinities of individual and combinations of nucleotides that remains to be done.

We now have much more information in the text about how we established the appropriate ratios of nucleotides for this high resolution work and explain the difficulties in obtaining structures with the other nucleotide substrates and effectors.

2) In the paper, as written, it is difficult for a non-specialist reader to discern what specifically is the advance reported in the paper, given the lower-resolution reconstruction of the hexameric α-subunit structure reported previously. There is some discussion of this in the concluding section of the paper, but the paper would be strengthened by bringing to the front a discussion of what is not known, and what conceptual advances are to be expected of the present work, beyond improving the resolution.

To bring more focus to the advances of this paper, we have modified the text throughout, with a major rewrite of the first paragraph of the Discussion as suggested. The reviewers are correct that the advances are more substantial than just an improvement in resolution, albeit a dramatic improvement. We agree that the complexity of the RNR enzymes requires an extra effort such that non-specialists can appreciate the findings. We hope our rewriting efforts now convey the advances more clearly.

Please note the other important points to address, shown below. Many of these points concern the clarity of the manuscript and the accessibility to a non-specialist reader.1) The paper is written in such a way that it is quite difficult for a non-specialist to appreciate the complex allosteric regulation of this enzyme, and to put the work in the context of previous results. The two kinds of allosteric sites (which are referred to throughout the paper) should be clearly explained, and the expectations as to which of these sites would be occupied (or not) in the hexameric structure should be discussed. The impact of the figures would be improved by accompanying schematic diagrams.

Indeed, RNR activity and specificity regulation is complicated and, despite having been discovered decades ago, remains incompletely understood. We attempted to clarify the schemes for activity and specificity regulation with the following modifications:

1) Previously Figure 1 conveyed only specificity regulation, we have modified this scheme to also include activity regulation and ATP.

2) To tie the regulatory schematic in Figure 1 with the binding sites on the α subunit, Figure 1 now lists the effectors and substrate nucleotides that bind at each site.

3) Figure 1 is now Figure 2 and shows the nucleotides at each site in spacefill and with labels. Furthermore the structures are labeled “active” and “inhibited”, and to emphasize that the alpha2beta2 model is just a docked alpha2 and beta2 structures, it has been relabeled alpha2+beta2 instead of alpha2beta2.

4) These graphical changes to Figure 1 are accompanied by changes to that figure legend.

5) We more clearly state in the text which effector sites are filled in the hexameric structure.

2) A schematic diagram should be provided that shows the allosteric modulators, and how they alter oligomeric states. Figure 1 shows specificity and activity effectors without explaining what they are.

We have rearranged the figure panels to create a figure that exclusively explains allosteric regulation of activity and the connection between effectors and enzyme oligomeric states. The Figure 2 legend now contains much more information about how allosteric activity regulation works for both *E. coli* RNR and human RNR.

3) Figure 1 shows a complicated feedback network in which different nucleotides affect the activity (and specificity?) of the enzyme. With the improved description of the enzyme, this diagram should be related to the known structural features (see the major concern noted above, which asks why the information on different effectors and substrates is limited).

We have remade Figure 1, generated a new figure (Figure 2), and substantially rewritten the text and figure legends to explain what is known and, more importantly, to explain why so little good structural data are available.

4) In Figure 1, an equilibrium between different oligomeric states is shown, but without explanation of how and why ATP shifts this equilibrium.

Figure 1 is now part of Figure 2. We have rewritten the figure legend to more fully describe what we do know about how ATP and dATP shift the equilibrium as well as describing what we do not know.

5) Why was a mixture of dATP and ATP used to prepare the cryoEM sample? There is one CDP molecule per subunit, but it is not described where this came from. The logic of the sample preparation should be explained.

We have added a new section to the Results in which the logic of sample preparation is explained in detail.

6) It would be interesting to know by how much the map resolution improved when 6-fold averaging was applied.

We collected several data sets prior to the one reported here and found that the resolution was 10 Å before applying symmetry and 5 Å after applying D3 symmetry. Confident that we could produce specimens that would give us a high-resolution structure we subsequently collected a final dataset on a Titan Krios with a K2 detector in counting mode and following the reconstruction procedure with D3 symmetry arrived at the reported near atomic-resolution structure. We did not examine the effects of applying different possible symmetries (C1, C2, or C3) during refinement.

7) Additional figures showing densities of residues in important regions should be provided, such as the cone interactions and dATP interaction, as well as the loop 2 conformation and substrate coordination.

We have added two supplemental videos to display the density. One video shows the cone domain and the other shows the loop 2 region. We agree that these density figures are important.